# Revealing Antibiotic Tolerance of the *Mycobacterium smegmatis* Xanthine/Uracil Permease Mutant Using Microfluidics and Single-Cell Analysis

**DOI:** 10.3390/antibiotics10070794

**Published:** 2021-06-29

**Authors:** Meltem Elitas, Neeraj Dhar, John D. McKinney

**Affiliations:** 1Faculty of Engineering and Natural Sciences, Sabanci University, 34956 Istanbul, Turkey; 2School of Life Sciences, École Polytechnique Fédérale de Lausanne (EPFL), 1015 Lausanne, Switzerland; neeraj.dhar@epfl.ch (N.D.); john.mckinney@epfl.ch (J.D.M.)

**Keywords:** antibiotics, conventional, microbiology, microfluidics, microscopy, *Mycobacterium smegmatis*, population, single cell

## Abstract

To reveal rare phenotypes in bacterial populations, conventional microbiology tools should be advanced to generate rapid, quantitative, accurate, and high-throughput data. The main drawbacks of widely used traditional methods for antibiotic studies include low sampling rate and averaging data for population measurements. To overcome these limitations, microfluidic-microscopy systems have great promise to produce quantitative single-cell data with high sampling rates. Using *Mycobacterium smegmatis* cells, we applied both conventional assays and a microfluidic-microscopy method to reveal the antibiotic tolerance mechanisms of wild-type and *msm2570::Tn* mutant cells. Our results revealed that the enhanced antibiotic tolerance mechanism of the *msm2570::Tn* mutant was due to the low number of lysed cells during the antibiotic exposure compared to wild-type cells. This is the first study to characterize the antibiotic tolerance phenotype of the *msm2570::Tn* mutant, which has a transposon insertion in the *msm2570* gene—encoding a putative xanthine/uracil permease, which functions in the uptake of nitrogen compounds during nitrogen limitation. The experimental results indicate that the *msm2570::Tn* mutant can be further interrogated to reveal antibiotic killing mechanisms, in particular, antibiotics that target cell wall integrity.

## 1. Introduction

Antibiotic tolerant bacteria, persisters, are among the key players in several refractory infectious diseases. Antibiotic tolerance might contribute to treatment failures, reoccurrence of infection or extremely long duration of antimicrobial therapy that might often lead to the evolution of antibiotic resistance [1,2,3,4]. Bacterial persisters are low-frequency, metastable phenotypic variants in clonal populations [4,5]. Several laboratories have used different models to investigate persisters. Various hypotheses have been examined to understand whether persister cells are dormant, killed slowly by antibiotics, divide slowly or do not divide in the presence of antibiotics, tolerate single or multiple antibiotic stresses, regrow upon antibiotic treatment, or live in higher antibiotic concentrations than the minimum inhibitory concentrations (MIC) on the class of antibiotics tested [5,6,7,8,9,10,11,12]. Very little is known about the single-cell kinetics and the underlying molecular mechanisms of these phenomena. Persistence has been mostly interrogated in model systems more than in naturally occurring persister cells using conventional methods. One of the biggest shortcomings of conventional tools is their inadequate detection limits to identify persisters or other rare cells in heterogeneous cell populations [9,10].

In the past decade, microfluidic and microscopy methods have arisen to overcome the limitations of traditional tools [11,12,13,14,15,16,17,18,19,20,21,22]. The advantages of microfluidic assays over macro-scale, traditional, batch culture methods include increased limit of detection, higher sampling rate, higher specificity, and compatibility with several imaging techniques [11,12,13,14,15,16,17,18,19,20,21,22]. Among these pioneering studies, Kim and co-workers presented that persister cells wake based on ribosome content [8]. Goormaghtigh and Van Melderen observed *Escherichia coli* (*E. coli*) cells that generated ofloxacin persisters at the single-cell level using a commercially available CellASIC ONIX B04A-03 Microfluidic Plate (Millipore, USA) and the CellASIC ONIX Microfluidic System [15]. They showed that persister cells did not constitute a slowly growing subpopulation as previously reported [10,14,15,16]. In their experimental model, persisters originated from metabolically active cells, which formed long, polynucleoid filaments and exhibited maximum SOS induction upon removal of ofloxacin. This study demonstrated that the persister subpopulation was heterogeneous. To investigate persistence, a case-by-case analysis was required at the single-cell level. In contrast, Pu et al. reported that dormancy was a passive defense mechanism of *E. coli* persisters that could be activated prior to enhanced efflux activity. They used single-cell, time-lapse microscopy to measure the fluorescence intensities of the cells in the microchannels [16]. Widels et al. first filtered persister cells, which exhibited β-lactam-induced filamentation, and then monitored awakening of *E. coli* cephalexin persisters using a microfluidic mother machine device [17]. Similarly, Manuse et al. sorted *E. coli* cells with low levels of energy-generating enzymes and then observed ATP levels using time-lapse images of the isolated cells in the mother machine device. They showed that cells with a low level of ATP survived in the presence of ampicillin [18]. 

On the other hand, the McKinney group mostly focused on mycobacterial persistence and widely investigated the behavior of *Mycobacterium smegmatis* (*M. smegmatis*), a model organism of *Mycobacterium tuberculosis* (*M. tuberculosis*, Mtb), in the presence of first-line anti-tuberculosis drugs [11,13,19,20]. Their ground-breaking studies showed that persistence was a dynamic behavior of antibiotic-stressed mycobacteria based on single-cell analysis using a microfluidic-microscopy system [13]. As a recent, innovative approach, Baron and co-workers developed a microfluidic acoustic-Raman platform to monitor, in real-time, the lipid concentration of *M. smegmatis* under the influence of isoniazid (INH). Their findings showed that INH increased the lipids in the mycobacteria, which might render isoniazid tolerance of the cells [21]. Moreover, Bielecka et al. used bioelectrospray technology to generate a three-dimensional (3D) microsphere system that incorporated primary human cells, *M. tuberculosis* cells, and type I collagen, where they could both mimic cellular aggregation and upregulate mycobacterial stress genes. Next, they combined the microspheres with a microfluidic platform to study real-time pharmacokinetic modeling [22]. More efficient experimental systems have been developed than the standard in vitro culture and animal models. With a better understanding of antibiotic tolerance or resistance mechanisms, the probability of discovering new antibiotics would be increased. 

In this study, we used a microfluidic chip integrated with a dialysis membrane, published in [13,19,20]. We elucidated an increased antibiotic tolerance mechanism for *M. smegmatis*. We performed both batch and microfluidic culturing for wild-type and the *msm2570::Tn* mutant of *M. smegmatis* cells. Our approach produced detailed quantitative data at higher sampling rates, which is more accurate and precise compared to conventional batch culture methods. To our knowledge, no one has previously characterized the phenotype of the *msm2570::Tn* transposon mutant that has an insertion in the *msm2570* gene encoding a putative xanthine/uracil permease. The persistency phenotypes of wild-type and the *msm2570::Tn* mutant cells have been shown to be in good agreement with those identified by traditional methods. This approach can be improved for automatization, and the generated data can be extended to high-throughput screens for deep-learning approaches.

## 2. Results

The *msm2570::Tn* transposon mutant was characterized using conventional assays including growth curves, drug-mediated killing, and stress-response assays. Upon understanding whether the phenotype of the *msm2570::Tn* transposon mutant was antibiotic-specific, microfluidic cell culture experiments were conducted using fluorescent time-lapse microscopy. We quantified the behavior of the *msm2570::Tn* transposon mutant at single-cell resolution using the acquired time-lapse videos.

### 2.1. Conventional Assays

The first approach was to determine the growth rate of the *msm2570::Tn* transposon mutant measuring culture turbidity (OD_600nm_) at the indicated time points, Figure 1. Growth of the *msm2570::Tn* transposon mutant in 7H9 liquid medium was indistinguishable from wild-type bacteria, Figure 1. Therefore, transposon insertion did not alter the proliferation rate of the cells.

To determine the impact of *msm2570* disruption on drug-mediated killing, we measured the killing rate of the *msm2570::Tn* mutant against the first-line and second-line antituberculosis drugs including isoniazid (INH), ethionamide (ETH), ethambutol (EMB), and rifampicin (RIF). *M. smegmatis* wild-type and *msm2570::Tn* transposon mutant strains were exposed to (a) INH 50 µg mL^−1^, (b) ETH 200 µg mL^−1^, (c) EMB 5 µg mL^−1^, or (d) RIF 200 µg mL^−1^, Figure 2. Serial dilutions of the treated cultures were plated at the indicated time points to determine the percent survival (CFU, colony formation units assay). Results are means ± standard errors from three independent cultures. Figure 2 illustrates that the *msm2570::Tn* transposon mutant cells had a higher tolerance to INH and EMB compared to wild-type cells.

To evaluate the INH sensitivity of the *msm2570::Tn* mutant, we determined the minimum inhibitory concentration (MIC) of INH using both agar and microdilution assays. The MIC values for INH were 3.125 ± 0.02 µg mL^−1^ and 6.250 ± 0.06 µg mL^−1^ for wild-type and the *msm2570::Tn* mutant, respectively. This result confirmed that the transposon insertion in the *msm2570::Tn* mutant did not make the mutants more sensitive or resistant to INH. Moreover, the drug concentrations used in the killing rate experiments were 10-fold higher than the MIC values, Figure 2 and Figure 3.

When the wild-type locus encoding the *msm2570* gene was inserted back into the mutant, it fully complemented the mutant. This confirmed that the transposon insertion was responsible for the INH-tolerance phenotype, and that another mutation did not occur. Upon restoring the wild-type phenotype by complementation and assessing antibiotic-specific killing rate, we tested the sensitivity of the *msm2570::Tn* transposon mutant to environmental stresses including 45 °C heat shock, pH 4.5 acidic conditions, 1% sodium dodecyl sulfate (SDS) detergent, and phosphate-buffered saline (PBS) nutrient starvation, Figure 4. 

The *msm2570::Tn* transposon mutant did not show increased sensitivity to any of these stresses (i.e., the mutant was equally sensitive as wild-type cells), Figure 4. 

### 2.2. Microfluidic-Microscopy Assay

We investigated the underlying dynamics of the enhanced persistency of the *msm2570::Tn* mutant by analyzing acquired single-cell resolution, time-lapse fluorescence videos. We used the microfluidic platform, which was explained in detail in [13] and shown in Figure 5. 

In this system, the polydimethylsiloxane (PDMS) microfluidic device allowed stable nutrient flow and rapid change in growth conditions of the bacteria, which were sandwiched in between a coverslip and the dialysis membrane Figure 5A,B. The device was assembled using two parallel polymethylmethacrylate (PMMA) plates and six screws mounted on an inverted, automated, time-lapse fluorescent microscope (Olympus IX75) equipped with a Hamamatsu camera (ORCA-AG CCD). Medium flow (25–30 µL/min) was controlled by a syringe pump. Figure 5C illustrates the acquired images using a 100x oil immersion objective for 0–24, 24–48, 48–64, and 64–72 h. Since wild-type *M. smegmatis* cells were expressing green fluorescent protein (GFP), the images were obtained using both phase and green-fluorescent channels (150 µs). 

Using the microfluidic-microscopy assay, we acquired image stacks of premixed (1:1), wild-type (GFP-expressing), and the *msm2570::Tn* mutant (red fluorescent protein (RFP), RFP-expressing) cells. We manually counted cytolysis and division events of cells for a 4-h sampling time using the ImageJ cell counter plugin. Figure 6 shows the total cell number and the numbers of divided and killed cells during the antibiotic treatment using the microfluidic-microscopy system. GraphPad Prism 4 was used for the visualization of the data. 

## 3. Discussion

Microfluidic devices and microscopy techniques have been widely used to reveal antibiotic tolerance mechanisms of bacteria, but only a few studies have reported their observations or findings with underlying genetic mechanisms [11,12,13,14,15,16,17,18,19,20,21,22]. We investigated the antibiotic tolerances of wild-type *M. smegmatis* and the *msm2570::Tn* mutant cells using a microfluidic bacteria culture platform, which is capable of providing stable flow and rapid medium change [13]. Using a microfluidic-microscopy assay, we elucidated that the *msm2570::Tn* mutant displayed higher tolerance to isoniazid in comparison to wild-type cells. The mutation in the *msm2570* gene is responsible for xanthine and uracil permease activity and is involved in uptake of nitrogen compounds when nitrogen is limited [23,24]. In this context, there was only Petridis and co-workers’ study that reported the *msm2570* gene in the list of differentially expressed genes that were involved in the uptake of nitrogen compounds during nitrogen limitation in *M. smegmatis*. Interestingly, they also revealed that the *msm2570* gene was differentially expressed under nitrogen stress conditions using a continuous culture system. However, when Williams et al. investigated the response of *M. smegmatis* cells to nitrogen stress using batch culture tools, the *msm2570* gene was not explicitly reported [25]. While both studies performed growth measurement and gene expression profiling, none has demonstrated the behavior of the *msm2570::Tn* mutant cells in the presence of antibiotics. Nonetheless, correlated results generated using our microfluidic culture and Petridis’ continuous culture chemostat demonstrated the limitations of batch culture methods [23].

Our study, for the first time, reports the importance of the *msm2570* gene in antibiotic stewardship. The *msm2570::Tn* mutant displayed the most specific hyper persistence to INH and enhanced survival to EMB, Figure 2 and Figure 6. INH and EMB are two of the first lines of treatment for tuberculosis [25,26]. Both INH and EMB interfere with the biosynthesis of the cell membrane in *M. smegmatis*. It can be speculated that the killing mechanisms of INH and EMB might be affected due to the altered cell membrane composition of the *msm2570::Tn* mutant [24]. Thus, further research is needed to reveal the role of the *msm2570* gene in the action mechanisms of antibiotics that target cell wall integrity in *M. smegmatis*. It is important to emphasize that the *msm2570::Tn* mutant and wild-type cells showed a similar proliferation rate in normal growth medium, Figure 1, and a comparable killing profile against ETB and RIF, Figure 2. Moreover, the *msm2570::Tn* transposon mutant survived slightly better than wild-type under detergent (SDS) stress. This result might be correlated with the antibiotic responses of the *msm2570::Tn* mutant, which only showed higher antibiotic tolerance to drugs that target cell membranes. However, for the other stresses including heat shock, acidic pH, and nutrient starvation, the mutant responded similarly to wild-type cells. 

When the antibiotic responses of wild-type and the *msm2570::Tn* mutant cells were interrogated using microfluidic-microscopy assays, the obtained data were in good agreement with those obtained by conventional methods. However, the microfluidic-microscopy system allowed long-term, real-time observation of single cells before and during antibiotic exposure. Therefore, it overcomes the limitations of batch culture assays. It provides more reliable growth measurements in comparison to optical density measurements, which rely on the turbidity of the cell population without eliminating the contribution of dead cells and cell debris. Notably, optical density measurements are lacking single-cell level data; they provide the mean value for the growth of the population assuming that a population always has a normal distribution. On the other hand, this method can generate rapid, quantitative, real-time data for automation and high-throughput readouts. 

Colony-Forming Unit (CFU) assays remain the gold standard for evaluating the effectiveness of antibiotics. To reveal the underlying drug killing mechanisms, accurate, real-time, and high-rate sampling of cell division and lysis data are required, which makes the CFU assay unfeasible for automation and digitalization. Our single-cell data obtained using the microfluidic-microscopy system were correlated with the CFU assay, Figure 2 and Figure 6. The CFU assay is limited to deciphering the viable cells that can form colonies [27,28]. Real-time monitoring of cells might address this problem and provide more accurate data with higher sampling frequencies. Moreover, the CFU assay provided the number of cells which can form colonies at 0, 24, 48, and 72 h, Figure 2. In our study, we evaluated the behavior of single cells in each image sequence with a four-hour sampling rate. Figure 6 shows the number of divided and lysed cells (Figure 6C,D) with the total cell numbers (Figure 6A,B). The cumulative behavior of wild-type *M. smegmatis* (Figure 6A) and the *msm2570::Tn mutant* (Figure 6B) cells followed a similar killing profile to that observed in batch culture (Figure 2A,C). Notably, the single-cell analysis made it possible to understand the behavior of the cells when the cell numbers are at a steady state between two consecutive time intervals in CFU assays. By monitoring the bacterial behavior in the presence of antibiotics at the single-cell level, the difference in their survival strategies can be elucidated as shown for wild-type cells (Figure 6C) and the *msm2570::Tn* mutant cells (Figure 6D). First, the cumulative cell number was increased both for wild-type and the *msm2570::Tn* mutant cells due to more division than lysis events. Next, the number of lysed cells was higher than the number of divided cells for both cell populations. However, compared with wild-type, the *msm2570::Tn* mutant exhibited less cell lysis. Consequently, the total cell number was higher for the *msm2570::Tn* mutant, Figure 6. Interestingly, single-cell analysis explained that the steady-state survival of wild-type cells between the time points of 20–48 h in the CFU assay (Figure 2) was due to an equal number of division and lysis events in the population (Figure 6C). Although this system can be widely applied to the administration of different antibiotics to different cell types, manually counting the number of divided cells and lysed cells was cumbersome and can still lead to inaccurate enumeration in single-cell analysis. Therefore, accurate and precise segmentation of mycobacterium cells is still a great challenge to overcome to make microfluidic-microscopy systems more effective. 

## 4. Materials and Methods

### 4.1. Growth Rate Measurements

The growth rate of the *M. smegmatis* cells in Middlebrook 7H9 medium (BD/Difco) at 37 °C, 3 g was determined by measuring time-dependent changes in optical density at 600 nm (OD_600_) (Thermo Scientific (Waltham, MA, USA) biomate 5). Middlebrook 7H9 medium contained 0.5% albumin, 0.2% glucose, 0.085% NaCl, 0.5% glycerol, and 0.05% Tween-80.

### 4.2. Minimum Inhibitaroy Concentratoin (MIC)

The MIC values of INH for the *msm2570::Tn* mutant were determined using both agar proportion and microdilution assays. For both methods, cells were grown to mid-log phase (OD_600_ of 0.5–1). For the agar proportion method, cells were diluted to an OD_600_ of 0.05 in fresh 7H9 medium. Then 10-fold serial dilutions were plated on Luria–Bertani (LB) agar solid plates containing various concentrations of INH (2-fold serial dilutions: 100–0.024). Plates were incubated at 37 °C, and the lowest concentration of drug required to detect minimum bacterial CFU was determined. For the microdilution method, cells were diluted to an OD_600_ of 0.005 in fresh 7H9 medium. Then, 2-fold serial dilutions of INH ranging from 100 µg mL^−1^ to 0.024 µg mL^−1^ was prepared with 3 mL of cultures that were aliquoted into 15 mL test tubes. As a control, one tube of culture was left drug-free, and one tube was fresh 7H9 medium as a blank. Tubes were incubated at 37 °C for 3 days. Turbidity of the cultures was read using OD_600_ absorbance on a TECAN plate reader. For the plate reader, 200 µL of cells was aliquoted from each tube into Costar 96-well transparent flat-bottom plates. When the measurements were done, the absorbance of the blank was subtracted from readouts from each of the wells. MIC was defined as the lowest concentration of drug required to detect minimum bacterial growth.

### 4.3. Killing Rate Measurements

The killing rate of the *msm2570::Tn* mutant was investigated using a conventional CFU assay. The wild-type control strain and the *msm2570::Tn* mutant cells were grown in 7H9 medium overnight at 37 °C ~3 g. When the cultures were in log phase (OD_600_ ≈ 0.5), the wild-type and mutant strains were diluted to an OD_600_ of 0.05 with standard 7H9 medium. At time zero, antibiotics were added to the diluted cell cultures. At 6, 24, 48, and 72 h, survival was assessed by withdrawing aliquots of the cultures, plating 10-fold serial dilutions on LB agar, and scoring colonies after 3–4 days at 37 °C. The killing rate measurements were performed for isoniazid (INH) 50 µg mL^−1^, rifampicin (RIF) 200 µg mL^−1^, ethionamide (ETH) 200 µg mL^−1^, and ethambutol (EMB) 5 µg mL^−1^. The drug concentrations used in our experiments were 10-fold higher than the MIC values.

### 4.4. Complementation Assay

The c2570F and c2570R primer pair was designed with the following sequences, respectively, GCT AGC ATG ACC ATT CCT TCT GCC and GAT ATC TCA CGA AAG TCG GTC GGA GAC. The *msm2570* gene was PCR-amplified using these primers with NheI and EcoRV restriction sites. The amplified gene was sub-cloned into PCR2.1 TOPO, checked by digestion, and sequenced to check for errors. The pND200_Strep integrative plasmid was ligated with the *msm2570* gene. The streptomycin-resistance encoding pND200_strep_*msm2570* plasmid was transformed into DH5α *E. coli* cells.

### 4.5. Sensitivity Assay

The sensitivity of the *msm2570::Tn* mutant to other stresses was assessed. First, thermal stress was tested at 45 °C for 6 h. Next, cells were exposed to 0.1% sodium dodecyl sulfate (SDS) detergent for 24 h. Then, nutrient starvation of the *msm2570::Tn* mutant was determined by incubation of cells in phosphate-buffered saline (PBS) for 24 h. Afterward, the *msm2570::Tn* mutant was cultured in medium with acidic conditions, pH 4.5, for 24 h. The 0, 6, and 24 h survival of the *msm2570::Tn* mutant was assessed by withdrawing aliquots of the cultures, plating 10-fold serial dilutions on JALB (Luria-Bertani) agar, and scoring colonies after 3–4 days at 37 °C.

### 4.6. Microfluidic-Microscopy Assay

The RFP-expressing *msm2570::Tn* mutant cells and GFP-expressing wild-type *M. smegmatis* cells were mixed at a 1:1 ratio. The mixture of the cells was sandwiched in between a coverslip (#1) and the semipermeable membrane (Spectrum Lab, MWCO:8), which is under the PDMS (Sylgard 184) microfluidic channels (50 µm × 50 µm) [10]. The microfluidic chip provided growth of the cells via continuous medium flow (25–35 µL min^−1^, silicone tubing ID/OD: 0.076/0.165 cm, HelixMark, syringe pump: World Precision Instruments (Sarasota, FL, USA)) and the possibility of medium switching for on-chip drug exposures. The design and operation of the microfluidic device were explained in detail [13]. The microfluidic device loaded with cells was mounted on the microscope inside a temperature-controlled chamber (37 °C). The microfluidic assay included three steps, 24-h growth of cells in 7H9 medium, 72 h drug exposure using 50 µg/mL INH (as applied in the CFU assay), and 24-h regrowth of the cells in the absence of INH.

### 4.7. Live-Cell Imaging and Single-Cell Analysis

Live-cell imaging was performed using an inverted, automated, time-lapse fluorescent microscope (Olympus IX75) equipped with a Hamamatsu ORCA-AG CCD camera. Using a 100× oil immersion objective (UPLFLN), images were acquired on phase and fluorescent channels (TRIS-red, GFP-green, exposure: 150 µs). Then, images were converted into time-lapse fluorescence movies for image analysis. We used the ImageJ cell counter plugin to count the number of divided and lysed cells for every 4-h time frame. Visualization of the data was performed in GraphPad Prism 4 software. 

### 4.8. Statistical Analysis

Student’s unpaired *t*-test (two-tailed) was used to assess the statistical significance of pairwise comparisons. *p* values < 0.05 were considered significant. *p* values were calculated using GraphPad Prism 4 software. 

## 5. Conclusions

In this study, we implemented both a microfluidic-microscopy method and batch culture assays to better understand the antibiotic tolerance behavior of wild-type and the *msm2570::Tn* mutant of *M. smegmatis* cells. Our findings were confirmed by conventional methods including optical density measurements and CFU assays. Moreover, microfluidic-microscopy techniques revealed that the enhanced antibiotic tolerance mechanism of the *msm2570::Tn* mutant was due to the low number of lysed cells during the antibiotic exposure compared with wild-type cells. This is the first study that characterized the phenotype of the *msm2570::Tn* transposon mutant that has a transposon insertion in the *msm2570* gene encoding a putative xanthine/uracil permease. Our technique produced quantitative data at higher sampling rates in comparison to traditional methods. Therefore, this technique can be conveniently improved for automation and digitalization processes. The first step of this process requires the development of rapid, precise, and accurate image segmentation tools for *M. smegmatis* cells. Segmentation of label-free mycobacterium cells by the naked eye is cumbersome and leads to inaccurate and imprecise results. Therefore, there is an urgent need for the development of rapid, robust, and effective deep learning approaches for the segmentation of label-free mycobacteria. 

## Figures and Tables

**Figure 1 antibiotics-10-00794-f001:**
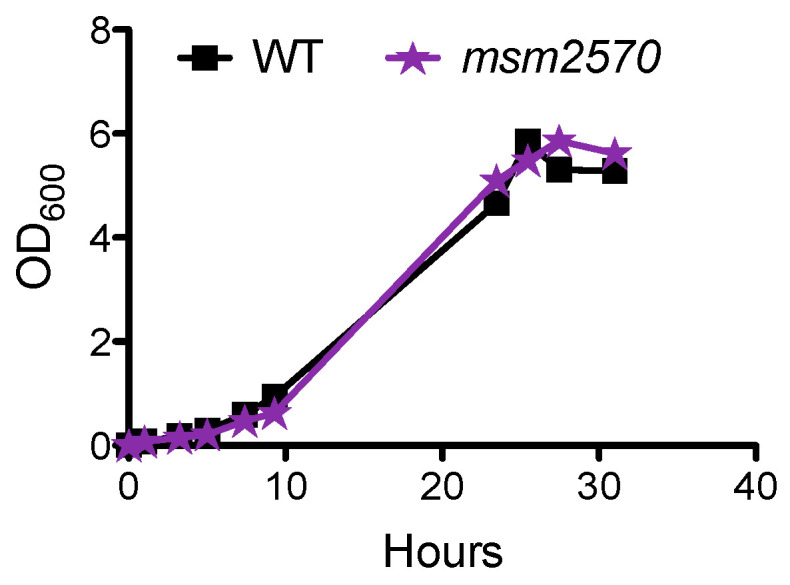
Growth of the *msm2570::Tn* mutant in 7H9 medium. Wild type (■), the *msm2570::Tn* mutant (⋆). Results are representative of at least two experiments.

**Figure 2 antibiotics-10-00794-f002:**
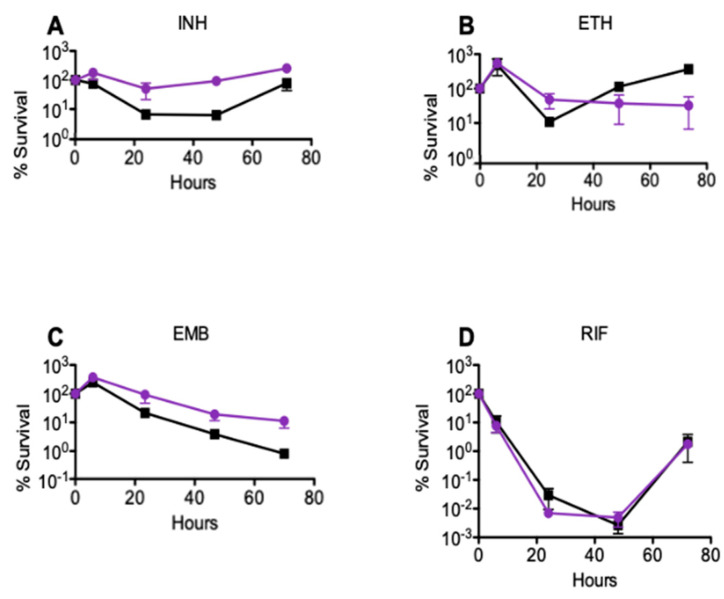
Response of the *msm2570::Tn* transposon mutant to antibiotics in batch cultures. *M. smegmatis* wild-type (■) and *msm2570::Tn* mutant (•) strains were treated with (**A**) INH 50 µg mL^−1^, (**B**) ETH 200 µg mL^−1^, (**C**) EMB 5 µg mL^−1^, (**D**) RIF 200 µg mL^−1^. Results are the means ± standard errors from three independent cultures.

**Figure 3 antibiotics-10-00794-f003:**
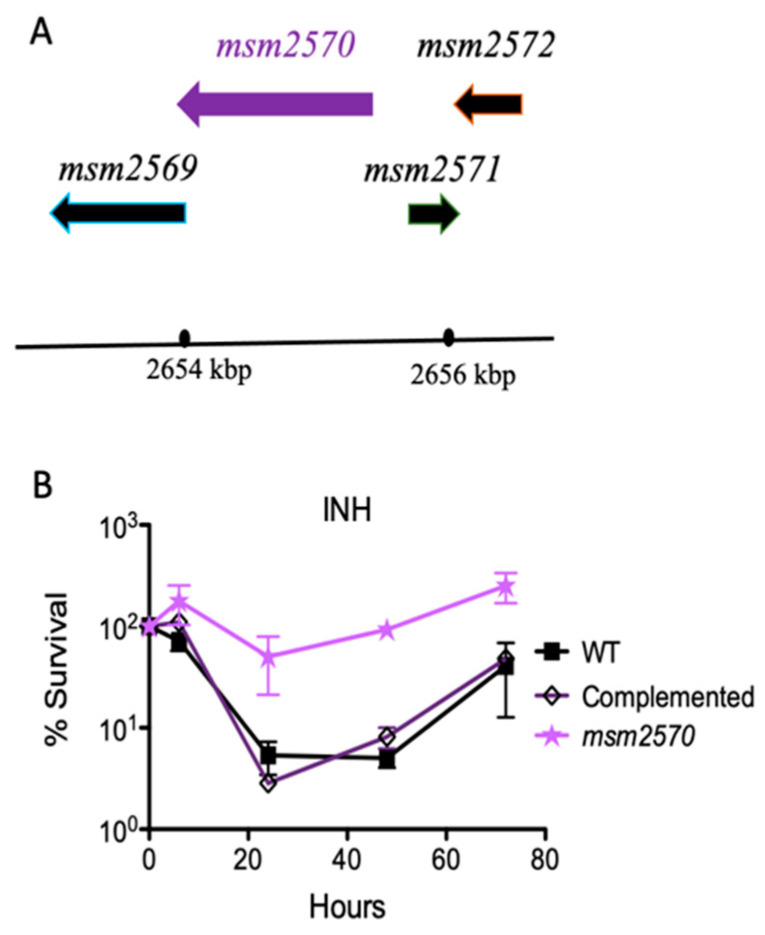
Complementation of the *msm2570::Tn* mutant. (**A**) Chromosomal locus encoding the *msm2570* gene in *M. smegmatis*. (**B**) INH killing assay for the complemented *msm2570::Tn* mutant in batch culture. *M. smegmatis* wild-type (■), the *msm2570::Tn* mutant (⋆), complemented strain (
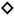
) strains were treated with INH 50 µg mL^−1^. Results are the means ± standard errors from three independent cultures.

**Figure 4 antibiotics-10-00794-f004:**
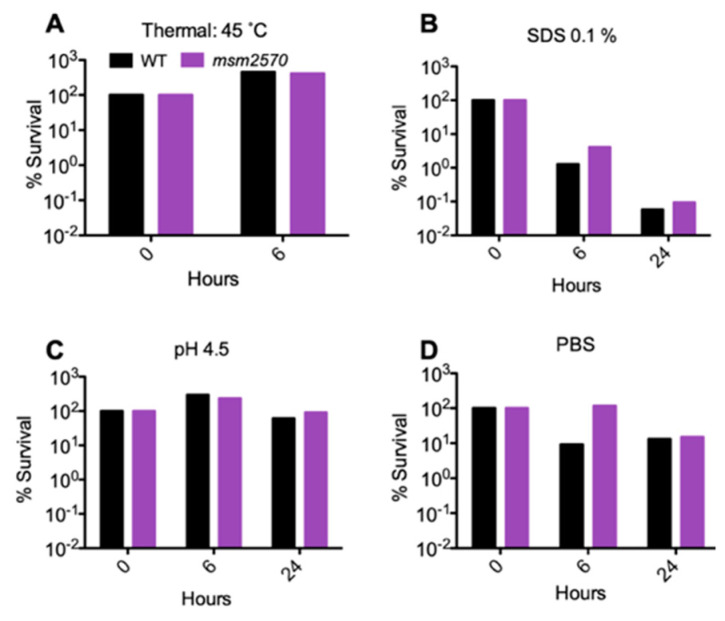
Sensitivity of the *msm2570::Tn* mutant and wild-type cells to thermal, SDS, pH, and PBS stresses. Percent survival of WT and the *msm2570* mutant at 45 °C (**A**), after incubation in 7H9 in the presence of 0.1% SDS (**B**), pH 4.5 (**C**), or PBS (**D**).

**Figure 5 antibiotics-10-00794-f005:**
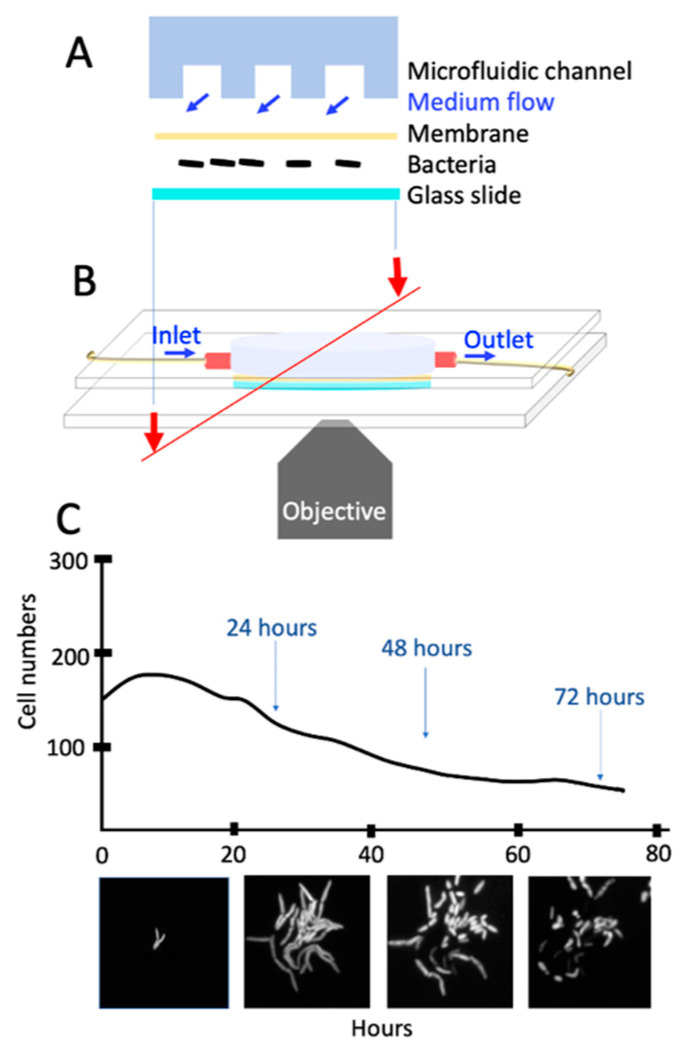
A single-cell assay using microfluidics and microscopy. (**A**) Cross-section of the microfluidic device with its components. (**B**) Microfluidic device on the microscope stage for live-cell fluorescence imaging. (**C**) View of the bacteria in the microfluidic device and phases of antibiotic exposure.

**Figure 6 antibiotics-10-00794-f006:**
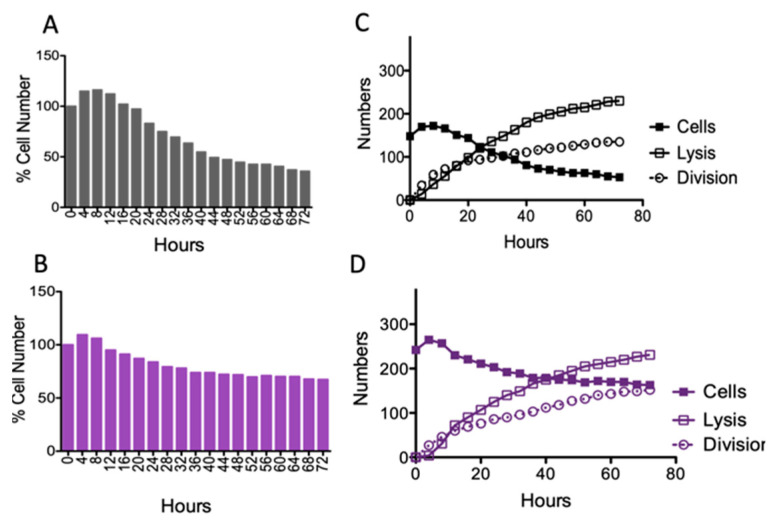
Quantitative single-cell analysis. The cumulative behavior of (**A**) wild-type *M. smegmatis* and (**B**) the *msm2570::Tn* mutant cells in the presence of 50 µg/mL INH exposure. The number of cytolyzed and divided cells are shown for (**C**) wild-type *M. smegmatis* and (**D**) the *msm2570::Tn* mutant cells.

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
