# Peer review of "Revealing Antibiotic Tolerance of the Mycobacterium smegmatis Xanthine/Uracil Permease Mutant Using Microfluidics and Single-Cell Analysis"

_antibiotics, 2021, doi:10.3390/antibiotics10070794_

Round 1

Reviewer 1 Report

Dear authors,

I want to highlight only some minor points:

  • Referring to figure 4b, authors state that msm2570::Tn mutant survived slightly better than WT in the presence of 0,1% SDS. In my opinion this better survival in not clear from the graph. It seems to me that the very little difference could fall within the experimental error, being the difference at 24hours, even smaller than at 6hours; or the grid on the y-axis isn’t clearly distinguishable to me. Authors could clarify this point, stating in the figure legend the experimental error of the measurements.
  • Authors address to the microfluidics devices and microscopy techniques as a method that can generate: “rapid, quantitative, real-time data for automation and high-throughput readouts” (line 237-238). My only concern is on the adjective rapid. This method doesn’t reduce investigation time respect CFU assay.
  • As concern the microfluidic measurement, is not stated the medium and the antibiotic concentration used, nether in Material and Methods nether in the legend of the figure 6.

Author Response

First, we would like to thank our editor and reviewers for their kind and supportive revision. We highly appreciated our reviewers’ valuable feedbacks.

Reviewer_1

  • Referring to figure 4b, authors state that msm2570::Tn mutant survived slightly better than WT in the presence of 0,1% SDS. In my opinion this better survival in not clear from the graph. It seems to me that the very little difference could fall within the experimental error, being the difference at 24hours, even smaller than at 6hours; or the grid on the y-axis isn’t clearly distinguishable to me. Authors could clarify this point, stating in the figure legend the experimental error of the measurements.

Author comments Reviewer1_C1: We completely agree with our reviewer, therefore we changed the term “msm2570::Tn mutant survived slightly better than WT in the presence of 0,1% SDS.”

Line 154-156: The msm2570::Tn transposon mutant did not show increased sensitivity to any of these stresses (i.e., the mutant was equally sensitive as wild-type cells), Figure 4.

  • Authors address to the microfluidics devices and microscopy techniques as a method that can generate: “rapid, quantitative, real-time data for automation and high-throughput readouts” (line 237-238). My only concern is on the adjective rapid. This method doesn’t reduce investigation time respect CFU assay.

Author comments Reviewer1_C2: Thank you very much for your careful revision, we agree with our reviewer and corrected the sentence as below:

Line 238-240: To reveal the underlying drug killing mechanisms, accurate, real-time, and high-rate sampling of cell division and lysis data are required, which makes CFU assay not feasible for automation and digitalization.

  • As concern the microfluidic measurement, is not stated the medium and the antibiotic concentration used, nether in Material and Methods nether in the legend of the figure 6.

Author comments Reviewer1_C3: We agree with our reviewer, it was our mistake not to state that the microfluidic measurements were performed in the presence of 50-µg/ml INH as we exposed our cells in the CFU assay.

  • Figure 6. Quantitative single-cell analysis. The cumulative behavior of (a) wild-type smegmatis and (b) the msm2570::Tn mutant cells in the presence of 50 µg/ml INH exposure. The number of cytolyzed and divided cells were for (c) wild-type M. smegmatis and (d) the msm2570::Tn mutant cells.

Material and Methods section:

Line 335-337: The microfluidic assay included three steps, 24-hour growth of cells in 7H9 medium, 72-h drug exposure using 50 µg/ml INH (as applied in the CFU assay), and 24-hour regrowth of the cells in the absence of INH.

Thank you very much for your careful revision and valuable suggestion, we greatly appreciated.

Reviewer 2 Report

minor points:

Some sentences are hard to read or incorrect. Two examples are presented below but some English corrections are required for small and multiple ( but nevertheless avoidable) mistakes across the manuscript.

1- Among these pioneering studies, Goormaghtigh and Van Melderen observed Esche- richia coli (E. coli) cells those generated ofloxacin persisters at single-cell level using a com- 50 mercially available CellASIC ONIX B04A-03 Microfluidic Plate (Millipore, USA) and the 51 CellASIC ONIX Microfluidic System [14].

R- Maybe “that” instead of “those”

2- “To our knowledge, none has been previously characterized 89 the phenotype of the msm2570::Tn transposon mutant that has an insertion in the msm2570…”

R- maybe “To our knowledge, no one has previously characterized 89 the phenotype of the msm2570::Tn transposon mutant that has an insertion in the msm2570…”

Major Points:

1- The authors claimed they "reveal antibiotic-tolerance mechanisms for wild type and the 353 msm2570::Tn mutant of M. smegmatis cells". This is far from the reality - sure there  appears to be a clear effect of the mutant on some antibiotic resistance but apart from that descriptive data there is no mechanism for antibiotic tolerance here. The authors state that single cell analysis has advantages versus classical methodology - that is true - but has been shown previously as stated in the introduction.

2- Really weird results with the WT and mutant in the % survival – there appears to be 100 % survival for the WT with INH, ETH, only rifampicin works. In order to see a difference between WT and the mutant you need to see some killing…. (figure 2.). You should find conditions were you see killing in the WT to start with -  otherwise what are you comparing?

3- “To evaluate INH sensitivity of the msm2570::Tn mutant, we determined the minimum 129 inhibitory concentration (MIC) of INH using both agar and microdilution assays. The MIC 130 values for INH were 3.125 μg ml-1 and 6.250 μg ml-1 for wild type and the msm2570::Tn 131 mutant, respectively. This result confirmed that transposon insertions of the msm2570::Tn 132 mutant did not make the mutants more sensitive or resistant to INH.”

Don’t understand here the authors – so the MIC doubles for INH? and the mutant makes no difference? what would be a difference?

Author Response

First, we would like to thank our editor and reviewers for their kind and supportive revision. We highly appreciated our reviewers’ valuable feedbacks.

Reviewer_2. Comments and Suggestions for Authors minor points:

Some sentences are hard to read or incorrect. Two examples are presented below but some English corrections are required for small and multiple ( but nevertheless avoidable) mistakes across the manuscript.

Thank you very much we revised the whole manuscript for English.

1- Among these pioneering studies, Goormaghtigh and Van Melderen observed Esche- richia coli (E. coli) cells those generated ofloxacin persisters at single-cell level using a com- 50 mercially available CellASIC ONIX B04A-03 Microfluidic Plate (Millipore, USA) and the 51 CellASIC ONIX Microfluidic System [14].

R- Maybe “that” instead of “those”

Author comments Reviewer2_C1: Thanks, corrected as follows:

Line 49-52:  Among these pioneering studies, Goormaghtigh and Van Melderen observed Escherichia coli (E. coli) cells that generated ofloxacin persisters at single-cell level using a commercially available CellASIC ONIX B04A-03 Microfluidic Plate (Millipore, USA) and the CellASIC ONIX Microfluidic System [14]. They showed that persister cells did not constitute

2- “To our knowledge, none has been previously characterized 89 the phenotype of the msm2570::Tn transposon mutant that has an insertion in the msm2570…”

R- maybe “To our knowledge, no one has previously characterized 89 the phenotype of the msm2570::Tn transposon mutant that has an insertion in the msm2570…”

Author comments Reviewer2_C2: Thank you very much, corrected as below.

Line 89-91: To our knowledge, no one has previously characterized the phenotype of the msm2570::Tn transposon mutant that has an insertion in the msm2570 gene encoding a putative xanthine/uracil permease.

Major Points:

1- The authors claimed they "reveal antibiotic-tolerance mechanisms for wild type and the 353 msm2570::Tn mutant of M. smegmatis cells". This is far from the reality - sure there  appears to be a clear effect of the mutant on some antibiotic resistance but apart from that descriptive data there is no mechanism for antibiotic tolerance here.

Author comments Reviewer2_C3: We toned down the sentence that our reviewer mentions as below. Here, we don’t have mechanistic details, but single cell analysis allowed us to capture dynamics of antibiotic tolerance through measurement of division and lysis rates.

Line 353-355: In this study, we implemented both a microfluidic-microscopy method and batch culture assays to better understand antibiotic-tolerance behavior of wild type and the msm2570::Tn mutant of M. smegmatis cells.

The reviewer states that “single cell analysis has advantages versus classical methodology - that is true - but has been shown previously as stated in the introduction”. Also, we stated in the introduction that we first used single cell analysis for understanding the behavior of the msm2570::Tn mutant of M. smegmatis in the presence of INH, which has not been studied before.

2- Really weird results with the WT and mutant in the % survival – there appears to be 100 % survival for the WT with INH, ETH, only rifampicin works. In order to see a difference between WT and the mutant yogu need to see some killing…. (figure 2.). You should find conditions were you see killing in the WT to start with -  otherwise what are you comparing? 

Author comments Reviewer2_C4: We think that our reviewer seems to have misinterpreted the data. There is clear killing of WT by INH. See 24h and 48h time point vs 0h. The rebound at 72 h is most likely due to resistant cells regrowing. Also at 24h and 48h clear difference in killing between WT and mutant. Also see the complementation experiment figure.

Figure 2. Response of the msm2570::Tn transposon mutant to antibiotics in batch cultures. M. smegmatis wild-type () and msm2570::Tn mutant (o) strains were treated with (a) INH 50 µg ml-1, (b) ETH 200 µg ml-1, (c) EMB 5 µg ml-1, (d) RIF 200 µg ml-1. Results are the means ± standard errors from three independent cultures.

Figure 3. Complementation of the msm2570::Tn mutant. (a) Chromosomal locus encoding the msm2570 gene in M. smegmatis. (b) INH killing assay for complemented msm2570::Tn mutant in batch culture. M. smegmatis wild-type (), the msm2570::Tn mutant (*), complemented strain () strains were treated with INH 50 µg ml-1. Results are the means ± standard errors from three independent cultures.

3- “To evaluate INH sensitivity of the msm2570::Tn mutant, we determined the minimum inhibitory concentration (MIC) of INH using both agar and microdilution assays. The MIC values for INH were 3.125 μg ml-1 and 6.250 μg ml-1 for wild type and the msm2570::Tn mutant, respectively. This result confirmed that transposon insertions of the msm2570::Tn mutant did not make the mutants more sensitive or resistant to INH. Don’t understand here the authors – so the MIC doubles for INH? and the mutant makes no difference? what would be a difference?

Author comments Reviewer2_C5: Again, we think that there seems to be small misunderstanding. The MIC shifts by 2-fold, but such shifts are very negligible and often within experimental variation. This cannot be classified as classical shift in MIC resulting in resistance which usually results in shift of MIC 5-10 fold or even higher. Also, the kill curves were performed at much higher fold MIC values, so such small shifts in MIC differences between the strains should be nullified.

We greatly appreciated our reviewers’ kind help and valubale time. Thank you very much for your careful revision.

Round 2

Reviewer 2 Report

The manuscript is basically the same. Despite the authors claim I do think figure 2 is weird. It would be appropriate to adjust the scale of each experiment to the data - what is the point of going as low as 10-3 in the INH experiment?

Moreover for the MIC determination - sure this could be just a precision problem - but how can the reader know data if there is no standard deviation or any idea of the dispersion of the mic results?

Author Response

First, we would like to thank our editor and reviewers for their kind and supportive revision. We highly appreciated our reviewers’ valuable feedbacks.

Reviewer_2. Comments and Suggestions for Authors minor points:

Comments and Suggestions for Author

Comment 1: The manuscript is basically the same. Despite the authors claim I do think figure 2 is weird. It would be appropriate to adjust the scale of each experiment to the data - what is the point of going as low as 10-3 in the INH experiment?

Response to Reviewer’s 1. comment: Thanks. As our reviewer asks, the figure 2 is revised as follows:

(Please see attachment)

Comment 2: Moreover for the MIC determination - sure this could be just a precision problem - but how can the reader know data if there is no standard deviation or any idea of the dispersion of the mic results?

Response to Reviewer’s 2. comment: Thank you, we agree with our reviewer and the standard deviations are included into the text.

Line 130-131: The MIC values for INH were 3.125±0.02 µg ml-1 and 6.250±0.06 µg ml-1 for wild type and the msm2570

 Thank you very much for our reviewer’s kind help and valuable time.

Round 3

Reviewer 2 Report

The authors inserted the standard deviations for the MIC results - not a big dispersion, in fact, showing that the MIC results are indeed different. If the authors understand that nevertheless the difference is not important that should be stated in the manuscript - it is not.

Contested figure was corrected.